

# Can we use atmospheric CO₂ measurements to verify emission trends reported by cities? Lessons from a six-year atmospheric inversion over Paris

Jinghui Lian[1,2], Thomas Lauvaux[3], Hervé Utard[1], François-Marie Bréon[2], Grégoire Broquet[2], Michel Ramonet[2], Olivier Laurent[2], Ivonne Albarus[1,2], Mali Chariot[2], Simone Kotthaus[4], Martial Haeffelin[4], Olivier Sanchez[5], Olivier Perrussel[5], Hugo Anne Denier van der Gon[6], Stijn Nicolaas Camiel Dellaert[6], and Philippe Ciais[2,7]

[1] Origins.earth, SUEZ Group, Tour CB21, 16 Place de l'Iris, 92040 Paris La Défense Cedex, France
[2] Laboratoire des Sciences du Climat et de l'Environnement (LSCE), IPSL, CEA-CNRS-UVSQ, Université Paris-Saclay, 91191 Gif sur Yvette Cedex, France
[3] Groupe de Spectrométrie Moléculaire et Atmosphérique (GSMA), Université de Reims-Champagne Ardenne, UMR CNRS 7331, Reims, France
[4] Institut Pierre Simon Laplace (IPSL), CNRS, École Polytechnique, Institut Polytechnique de Paris, 91128 Palaiseau Cedex, France
[5] AirParif, 7 rue Crillon, Paris, France
[6] Department of Climate, Air and Sustainability, TNO, P.O. Box 80015, 3508 TA Utrecht, Netherlands
[7] Climate and Atmosphere Research Center (CARE-C), The Cyprus Institute, 20 Konstantinou Kavafi Street, 2121, Nicosia, Cyprus

*Correspondence to:* Jinghui Lian (jinghui.lian@suez.com)

**Abstract**. Existing CO₂ emissions reported by city inventories usually lag real-time by a year or more and are prone to large uncertainties. This study responds to the growing need for timely and precise estimation of urban CO₂ emissions to support the present and future mitigation measures and policies. We focus on the Paris metropolitan area, the largest urban region in the European Union and the city with the densest atmospheric CO₂ observation network in Europe. We performed long-term atmospheric inversions to quantify the citywide CO₂ emissions, both fossil fuel and biogenic sources and sinks, over six years (2016-2021) using a Bayesian inverse modeling system. Our inversion framework benefits from a novel near-real-time hourly fossil fuel CO₂ emission inventory (Origins.earth) at 1 km spatial resolution. In addition to the mid-afternoon observations, we attempt to assimilate morning CO₂ concentrations based on the ability of the WRF-Chem transport model to simulate atmospheric boundary layer dynamics constrained by observed layer heights. Our results show a long-term decreasing trend by around 2% per year in annual CO₂ emissions over the Paris region. The impact of COVID-19 pandemic led to a 13%±1% reduction in annual fossil fuel CO₂ emissions in 2020 with respect to 2019. Then, annual emissions increased by 5.2% from 32.6±2.2 MtCO₂ in 2020 to 34.3±2.3 MtCO₂ in 2021. Based on a combination of up-to-date inventories, high-resolution atmospheric modeling, and high-precision observations, our current capacity could deliver near real-time CO₂ emission estimates at city scale in less than a month, and the results agree within 10% with independent estimates from multiple city-scale inventories.

## 1 Introduction

Most countries have actively committed to the 2015 Paris Agreement to limit global warming to well below 2°C, preferably 1.5°C, compared to pre-industrial levels. To achieve this goal, governments have pledged to implement stringent climate actions to reduce their national emissions with the ultimate objective of reaching climate neutrality by 2050. Cities account for more than 70% of annual global fossil fuel CO₂ emissions and thus are key areas for mitigating CO₂ emissions (Seto et al., 2014). To date, many metropolitan areas have pledged and began to implement policies to achieve net-zero emissions (e.g., C40 city, GCoM). The choice



of mitigation actions, for cost-effectiveness and to maximize emission reduction impact, depends mainly on the qualitative and quantitative understanding of urban emission sources with temporal-spatial details to understand evolving emission trends (Lauvaux et al., 2020; Mueller et al., 2021). However, the bottom-up carbon emissions based on public protocols for cities are prone to large uncertainties (Gurney et al., 2021). High-resolution gridded $CO_2$ emission inventories, e.g., the Hestia dataset for

some US cities (Gurney et al., 2019) or the LAEI dataset for Greater London (Greater London Authority, 2021), could provide a detailed description of emissions from urban domains. This approach relies on a collection of extensive activity data and emission factors, and thus can be labor-intensive and time-consuming, especially for doing regular updates. Recently, Carbon Monitor Cities has been developed to provide near-real-time city-level $CO_2$ emissions for 1500 global cities from 2019 to 2021 (Huo et al., 2022). The quantification of greenhouse gas (GHG) emissions from atmospheric measurements offers accounting complementary to the

conventional bottom-up approach (Ciais, 2010). These methods combine atmospheric measurements with bottom-up inventories through atmospheric inversion techniques (Tarantola, 2005). The scientific capabilities evolve rapidly with the deployment of dense networks and increasing model performances (Davis et al., 2017; Deng et al., 2017; Turner et al., 2020). The robustness of the inversion and the derived emission estimates need to be evaluated over periods of several months and years to check the stability and relevance of the seasonal cycle and of the inter-annual variability. For example, Mc Kain et al. (2012) indicated that their

transport model showed a poor performance in modeling urban sites, so that only relative changes in the emission estimates were considered relevant. To our knowledge, only three relatively long time series of estimates of city emissions based on atmospheric inversion systems have been published, covering a period over one to three years for the cities of Paris, Boston and Indianapolis (Staufer et al., 2016; Sargent et al., 2018; Lauvaux et al. 2020).

The Paris region, known as "Île-de-France" (IdF), is the highest populated and most economically active French region. Covering

only 2% of the French territory, it has around 18% of the French population (12.2 over 67.8 million inhabitants), and produces 31% of the national GDP and 10% of the human-caused GHG emissions of France (source: AirParif https://www.airparif.asso.fr/en/monitor-pollution/emissions; CITEPA https://www.citepa.org/fr/2022-co2e/). Paris is one of the most active cities in tackling climate change and part of the C40 City consortium. The first Paris Climate Plan was adopted in 2007 and targeted a 25% reduction in GHG emissions by 2020 with respect to 2004 levels. Paris city also has an ambitious 2020-2030

action plan which targets a 50% decrease in local direct GHG emissions (Scope 1) compared to 2004 levels (Le Plan Climat de Paris, 2020). Regarding atmospheric $CO_2$ monitoring capability, Paris is an important pilot city with the densest and most comprehensive atmospheric $CO_2$ measurements in Europe (e.g., Lopez et al., 2013; Xueref-Remy et al., 2018; Vogel et al., 2019; Lian et al., 2019). The Parisian ground-based network, whose first measurements date back to 2010, has grown from the initial three to the current seven high-precision continuous in-situ $CO_2$ monitoring stations. Over the past years, a series of studies have

attempted to analyze the spatial-temporal variations of $CO_2$ concentrations over the Paris region and to monitor fossil fuel $CO_2$ emissions through an atmospheric inversion technique (Bréon et al., 2015; Wu et al., 2016; Staufer et al., 2016). Lian et al. (2022) estimated $CO_2$ emission reductions during COVID-19 confinements in Paris, which demonstrated the capability of the urban atmospheric monitoring system to identify significant emission changes (>20%) at short-term monthly timescales.

This study performs the first long-term atmospheric $CO_2$ inversions over the Paris metropolitan area and compares it to multiple

city-scale inventories. It aims at assessing the ability and robustness of the inversion to track absolute urban $CO_2$ emission levels and its relative change over multiple years. The six years (2016-2021) continuous $CO_2$ measurements in Paris now provide sufficient information to investigate the variations in $CO_2$ emissions at different time scales (daily, seasonal and interannual) across an urban area. In addition to the mid-afternoon $CO_2$ concentration measurements that are commonly used for inversions, we also explore the potential for assimilating morning $CO_2$ data, taking into account the performance of the Weather Research and

Forecasting Model coupled with a chemistry transport model (WRF-Chem, Grell et al., 2005) in capturing the evolution of the



atmospheric boundary layer (ABL) dynamics. The height of the ABL is the main driver for uncertainties when assessing emissions from concentrations. This paper is organized as follows: Section 2 provides details of the city-scale Bayesian inversion methodology. Section 3 shows the variations in $CO_2$ emissions at different time scales. Section 4 summarizes the main conclusions and perspectives for further research.

## 2 Methods

A city-scale Bayesian inversion was conducted to quantify $CO_2$ emissions over a 6-year period spanning January 2016 to December 2021. The inversion system is based on atmospheric $CO_2$ measurements at seven in-situ stations combined with meteorological measurements, the WRF-Chem transport model run at 1 km × 1 km horizontal resolution (Lian et al., 2021), a near real-time fossil fuel $CO_2$ inventory produced by Origins.earth and the biogenic $CO_2$ fluxes simulated by the Vegetation Photosynthesis and Respiration Model (VPRM) included in WRF-Chem (Mahadevan et al., 2008). The main principle of the Bayesian atmospheric inversion presented in Lian et al. (2022) is to optimize the 6-hour mean fossil fuel $CO_2$ emission budgets of the Greater Paris region (Figure 1) over four time windows per day (0:00-5:00, 6:00-11:00, 12:00-17:00, 18:00-23:00 UTC). The approach assimilates atmospheric $CO_2$ concentration differences between pairs of stations located upwind and downwind of the city to decrease the uncertainties caused by the transport of remote and natural fluxes outside the urban area. Details regarding the inversion system setup are described in Lian et al. (2022) and outlined briefly below.

### 2.1 $CO_2$ measurement network

The seven stations are equipped with high-precision cavity ring-down spectroscopy (CRDS) $CO_2$ analyzers, together with an automated data processing and quality control system. $CO_2$ observations are calibrated every 1 to 6 months with standards traceable to the WMO $CO_2$ X2019 calibration scales (Hall et al., 2021). These stations are distributed roughly along a northeast-southwest axis of the Paris urban area which coincides with the predominant wind directions (Figure 1).

Figures S1 and S2 show the daily and monthly average daytime (8-17 UTC) $CO_2$ concentrations at each in situ station from 2016 to 2021, respectively and in addition Figure S2 also shows the simulated background $CO_2$ concentration at SAC station using the CAMS $CO_2$ data set as boundary and initial inputs for WRF-Chem. The atmospheric background $CO_2$ concentrations have been steadily rising over the past 6 years, primarily attributed to global human activities. Generally, the average $CO_2$ concentrations across the network vary seasonally between 390 and 450 ppm. They are mainly driven by the atmospheric transport, the $CO_2$ biospheric cycle, and the proximity to the urban anthropogenic $CO_2$ emission sources. The interannual $CO_2$ variations depend primarily on the year-to-year variations in synoptic weather conditions, air temperature and the associated emissions. For example, the notably high $CO_2$ concentrations observed near the surface during the 2016/17 winter were caused by stagnant, often stable atmospheric stratification associated with cold and dry air masses and low ventilation weather conditions over the north of France (Bulletin Climatique Météo-France, 2016 and 2017).

The gradients of $CO_2$ concentrations between the downwind and upwind stations are linked to the emissions within the Paris urban area. The $CO_2$ concentrations are significantly higher at the two urban stations CDS and JUS than those at peri-urban sites across all seasons (Figure S2). The magnitude in $CO_2$ gradients between urban and suburban areas is around 5~10 ppm in summer, increasing to 20~30 ppm during the winter months as a result of more stable atmospheric conditions (lower vertical dispersion and shallow ABLH) combined with high emissions from residential heating. The citywide $CO_2$ gradients across the Paris agglomerations have shown their potential in previous inversion studies to estimate the city-scale $CO_2$ emissions (Bréon et al., 2015; Staufer et al., 2016).



## 2.2 Origins.earth inventory

Fossil fuel $CO_2$ a priori fluxes used in this study are taken from a patent pending high-resolution inventory produced by Origins.earth (https://www.origins.earth/) over the IdF region, in combination with the global Open-source Data Inventory for Anthropogenic $CO_2$ (ODIAC) product (Oda et al., 2019) in surrounding areas. The Origins.earth bottom-up inventory is a gridded
map of fossil fuel $CO_2$ emissions within a rectangular (on a latitude and longitude grid) domain which encompasses most of the IdF region (Figure 1) at 1 km × 1 km spatial and hourly temporal resolution. It provides the Scope 1 $CO_2$ emissions from the year 2018 until the present time. For the simulation period from 2016 to 2017, we used $CO_2$ emissions from the Origins.earth inventory for the year 2018 as the WRF-Chem model inputs. The Origins.earth inventory includes more than 60 source types for carbon emission activities. These types of activities are grouped into six activity sectors (transportation, residential, tertiary, industry
including cement, energy, and waste). The inventory compilation method is outlined in SI Appendix (Text S1).

The IdF is a large urban region with estimated annual fossil fuel $CO_2$ emissions exceeding 30 $MtCO_2$ per year, dominated by traffic and residential sectors. The spatial distributions of the emissions are shown in Figure 1. According to the Origins.earth inventory, the annual emission budgets of the residential sector are 11.0, 10.9, 10.1 and 11.4 $MtCO_2$, representing 32%, 34%, 35% and 38% of total emissions for 2018, 2019, 2020 and 2021 respectively. For the traffic sector averages are 12.3 $MtCO_2$ (36%), 10.9 $MtCO_2$
(33%), 8.7 $MtCO_2$ (30%) and 8.7 $MtCO_2$ (29%) from 2018 to 2021 (Figure S3b). Figure S3a shows the daily fossil fuel $CO_2$ emissions by sector and their respective proportions from 2018 to 2021. Generally, the temporal variations of $CO_2$ emissions from the building sector show a large seasonal cycle, mainly related to the heating demand that is linearly driven by the variations of air temperature below a threshold of $\approx 19°C$. Emissions from the tertiary, industry and energy have a relatively flat seasonal variation. In this study, the inverse emissions are compared to independent estimates from different inventories and a published study. These
include (i) TNO-GHGco inventory at a resolution of 1/10º×1/20º (lon×lat) (~6 km×6 km) for the years 2005-2020 (Denier van der Gon et al., 2021), (ii) TNO-GHGco inventory at a resolution of 1/60º×1/120º (lon×lat) (~1 km×1 km) for the years 2015, 2017 and 2018 (Dellaert et al., 2019), (iii) AirParif inventory developed by the local official air quality agency (https://www.airparif.asso.fr/en/) at a 1 km × 1 km resolution for the years 2005, 2010, 2015 and 2018, (iv) the city-level $CO_2$ emissions from the Carbon Monitor Cities dataset (https://cities.carbonmonitor.org/, Huo et al., 2022) for the years 2019-2021, (v)
Staufer et al. (2016) reported a full year (August 2010-July 2011) estimate of the IdF region fossil fuel $CO_2$ emissions by assimilating citywide $CO_2$ measurements from a sparse network of three stations with the inversion methodology. Here, we consider that these multiple emission estimates allow for a cross-validation as they were developed by several research groups using distinct data sources, methods and protocols.

## 2.3 Adding morning $CO_2$ data and ABL height selection

The accuracy of atmospheric inversion results depends to a large extent on the quality of the atmospheric transport model. The major uncertainties in $CO_2$ modeling are related to model errors in horizontal wind and vertical mixing within the atmospheric boundary layer (Kretschmer et al., 2012). A stringent data selection of $CO_2$ concentrations to be assimilated for the inversion is generally applied (e.g., Staufer et al., 2016; Wu et al., 2016; Lauvaux et al., 2020). The aim of the data selection is to rule out observations that are difficult to simulate accurately with the transport model. Previous studies (Staufer et al., 2016; Lian et al.,
2022) assimilated only afternoon $CO_2$ data from 12 to 17 UTC during which the atmospheric boundary layer is expected to be well developed with a constant height. Moreover, the wind speeds at downwind stations are imposed to be higher than 3 m/s to minimize possible contaminations from local sources of $CO_2$ emissions near the measurement sites that cannot be reproduced by the model. However, the selection of a limited number of suitable data produces more uncertain estimates over periods when no observations are selected. The high sensitivity of the inverse emissions to the diurnal variations of prior emissions also highlights the limitations



induced by assimilating data only during the afternoon. Therefore, we attempt to include morning $CO_2$ concentration data in this study. Given that atmospheric models have difficulties in correctly reproducing the mixing processes under stable conditions and when the boundary layer develops in the morning, caution is required when assimilating morning $CO_2$ concentrations. We assumed that the ABL height could provide a good diagnostic to assess the vertical mixing and dilution associated with turbulence near the surface. We thus first evaluated the WRF-Chem simulated ABL heights against observations at the SIRTA station (Haeffelin et al., 2005) located about 20 km southwest of the Paris center (Figure 1). The modeled ABL heights were diagnosed from the potential temperature using the 1.5-theta-increase method (Nielsen-Gammon et al., 2008), while the aerosol-based STRATfinder algorithm (Kotthaus et al., 2020) was applied to derive ABL heights from attenuated backscatter profiles measured with an automatic lidar ceilometer (Lufft CHM15k).

Figure S4 shows hourly ABL heights derived from both measurements and simulations from 2016 to 2021 for morning (8-11 UTC) and afternoon (12-17 UTC) periods, respectively. In general, the model captures reasonably well the ABL heights throughout the year with a fair correlation coefficient (>0.6). However, the simulated ABL heights are, on average, higher than the measurements with biases of 80~130m. This could reflect an overestimation of the atmospheric instability in the lowest atmospheric layer by WRF or may in fact indicate that vertical dilution of atmospheric tracers (here observed aerosol profiles) may lag behind the

thermodynamic evolution of the ABL morning development (Text S2). Particularly large relative discrepancies are detected during the winter and morning periods associated with stable atmospheric conditions. This model-observation comparison of ABL heights allows us to revise the recent selection of $CO_2$ concentration measurements that can be assimilated into the inversion system as used in Lian et al. (2022) and extend it to the morning period. The revision consists primarily of adding two additional selection criteria concerning the morning data between 8-11 UTC. First, we discard the data when the relative error of the modeled ABL

heights against observations is larger than 80%. This threshold was set based on the distributions of the relative errors in ABL shown in Figure S4. Second, we apply a tighter filtering for the morning data (8-11 UTC) by increasing the minimum wind speed threshold to 5 m/s compared to the 3m/s used for the afternoon.

In addition to the reference inversion (Text S3), we also conducted a series of sensitivity tests to investigate how the inverse estimates respond to changes in various setups of the inversion system (Table S1). This inversion ensemble was composed of 5

tests of the selection criteria of the assimilated $CO_2$ observations, and 5 tests of the uncertainties and the temporal correlations of the prior emissions.

## 3 Results

### 3.1 Daily emission estimates

Statistical comparisons between the assimilated modeled and measured $CO_2$ concentration gradients before and after flux

optimizations are shown in Figure S5. In general, the inversion leads to an overall improvement in the representation of observations, both for the morning period (8-11 UTC) (Figure S5c) and the afternoon period (12-17 UTC) (Figure S5d). The mean bias errors (MBE) are reduced from -0.85 to -0.29 ppm in the morning and from -1.16 to -0.55 ppm in the afternoon after the inversion. The selected morning $CO_2$ gradients correspond to ~24.5% (5168 over 21091) of the total assimilated observation gradients.

Figure 2 compares the estimates from the reference inversion, which assimilates the daytime (8-17 UTC) $CO_2$ concentration gradients, to the estimates from the two sensitivity tests that only use morning (8-11 UTC) or afternoon (12-17 UTC) $CO_2$ data respectively. Here, we mainly focus on the Greater Paris region (Figure 1) where the fossil fuel $CO_2$ emissions can be well constrained from our observation network. Figure 2 shows the average prior and posterior fossil fuel $CO_2$ emission estimates



together with their associated uncertainties over the 6-year period spanning January 2016 to December 2022, for the four 6 h time windows and every day of the week. The time series of the daily prior and posterior fossil fuel $CO_2$ emissions are shown in Figure 3. The reference inversion (Figure 2b) mainly imposes a direct constraint on the fossil fuel fluxes for the two 6 h windows of the day (6-11 UTC and 12-17 UTC), while having little constraint on nighttime emissions (0-5 UTC and 18-23 UTC). This is because

the assimilated daytime $CO_2$ concentrations are mainly sensitive to the emissions from the morning and afternoon periods. The inversion increases the average posterior emissions per day of the week by around 9~16% and 13~23% with uncertainty reductions of 14~21% and 15~21% for the 6-11 UTC and 12-17 UTC time windows respectively. The assimilation of only afternoon $CO_2$ data (Figure 2d) has similar retrieved emissions but with a smaller uncertainty reduction especially for the 6-11 UTC period compared to the reference one which assimilates morning data. Even though we assimilate a relatively small number of morning

$CO_2$ data (Figure 2c), such a change still leads to a 11~16% uncertainty reduction compared to a priori of the morning fossil fuel emissions (Figure 2c).

The inversion on average reduces the uncertainty in the daily fossil fuel $CO_2$ emission estimates (Figure 3a) from ~31% (prior) down to ~24%±4.5% (posterior) (a range of 2.5%~11.5% uncertainty reduction) over the Greater Paris region. On the contrary, $CO_2$ emissions over the rest of IdF region (Figure S6) after the inversion remain close to their prior values due to insufficient

observational constraints. The inverse $CO_2$ emissions show a fairly good agreement with the prior estimate with a Pearson correlation coefficient (R) of 0.91, which indicates that the Origins.earth near-real-time inventory could already capture relatively well some timely emission variations that are closely related to meteorological effects and human activities. For instance, the lower emissions in February 2020 appears to be mainly due to weather conditions, with a mild winter leading to lower demand for heating (Météo-France climate bulletin, 2020). A decrease of the daily $CO_2$ emissions is observed in July and August due to a reduction in

traffic emissions during the summer holidays. It can also be seen that the daily fossil fuel $CO_2$ emissions dropped by more than 30% in the second half of March 2020, when the strict COVID-19 lockdown measures were adopted. In general, the inverse $CO_2$ emissions exhibit a larger daily variability compared to the prior. On the one hand, the temporal profiles used in the inventory rely on some degree of interpolation, proxy data and theoretical assumptions that can smooth out some temporal variability. On the other hand, the posterior daily variability represents both the actual variability in emissions caused by human activities and the

sources of uncertainty in the inverse modeling system, especially the transport model errors.

## 3.2 Monthly emission estimates

Figure 4a shows the posterior estimate of the monthly fossil fuel $CO_2$ emissions over the Greater Paris region derived through a temporal aggregation of the four 6 h period results. Results for the rest of IdF region are given in Figure S7. The inversion tends to increase the annual fossil fuel emissions by 2~6% with respect to the prior estimates for each of the years 2016 to 2021. It

demonstrates the consistency of the measurement constraint on inverse fossil fuel $CO_2$ emissions over time. It is important to note that the inversions for the years 2016 and 2017 were carried out with the prior emissions from the 2018 Origins.earth inventory, making the inverse emission changes mainly rely on the assimilated $CO_2$ observations. With the same prior emissions for 2017 and 2018, the inversion leads to an emission reduction by 0.6% over the Greater Paris region from 2017 to 2018, which is consistent with the general trend towards a decrease in emission.

Figure 4b shows the relative difference between the posterior and prior fossil fuel $CO_2$ emissions corresponding to the different inversion setups (e.g., prior errors and correlation length). It is worth noting that the optimized monthly emissions are very similar across the six years from 2016 to 2021. More precisely, the inversion consistently points to an average 10% increase in fossil fuel emissions in winter months compared to the prior (January to March). Further, the inverse emissions show significantly higher values (>20%) with respect to the prior during the spring period (especially in May and June). In contrast, minor differences





between the prior and posterior estimates are observed for the rest of the year. The ensemble of posterior fluxes, as represented by the spread of the minimum and maximum values among multiple inversions, provide an estimate of the variability introduced by a wide range of inversion setups. Even though these sensitivity tests may not represent the full potential range of uncertainties in the posterior fluxes, they still give us a certain degree of confidence in the inversion results because these variabilities are mostly

smaller than the posterior-prior emission differences and the emissions uncertainties.

We then conducted a series of analyses to investigate the potential explanations for the adjustments to the prior fluxes made by the inversion. First, we infer that the seasonal variability in flux adjustments is not caused by biases in the atmospheric transport model. This is because the model-observation misfits in wind speed exhibit a relatively constant pattern throughout a year without a pronounced seasonality (Figure S8). Second, the 10% increase in fossil fuel $CO_2$ emissions during the winter months (January to

March) could be due to an underestimation of the residential emissions in the Origins.earth inventory. The temporal profiles of the Origins.earth residential emissions are based on a proxy quantity derived from real-time domestic gas consumption data (Text S4). Whereas in reality, this seasonal cycle in the prior emissions may underestimate wintertime energy demands from other fuel types, especially for the petroleum and wood burning in the suburban residential areas (Figure S9). Notably, the estimated monthly emissions are ~20% larger compared to the prior estimates from April to June. Part of the adjustments of emissions could be due

to an underestimation of the biogenic fluxes from the VPRM model (Text S5). Figure S11 shows an obvious discrepancy between the VPRM simulated hourly biogenic flux (net ecosystem exchange, NEE) and the eddy flux measurements, especially during the growing season, at the Grignon cropland station located at around 40 km west of Paris center (Figure S10). The model-observation comparison of the vertical differences in $CO_2$ concentrations at the SAC station (15 m and 100 m above ground level) shows that the contribution of the nighttime biogenic respiration to the $CO_2$ concentration could also be a potential source of modeling errors

during the growing season (Lian et al., 2021).

### 3.3 Annual emission estimates

Verification of annual emission totals and tracking emission trends over time using multiple methods constitutes an indication of the reliability of inverse emissions estimates. Moreover, given the potential impact of atmospheric transport errors on the emission estimates, and assuming that the model errors do not change statistically over time, the emission trends are expected to be less

sensitive to model biases than the emissions estimates. The inter-annual variations are a critical metric to assess the effectiveness of the $CO_2$ mitigation efforts. Figure 5 shows the multiyear trend in the annual total fossil fuel $CO_2$ emissions over the IdF region for the period 2005-2021. The prior and inverse emissions are compared to multiple inventory datasets. The interannual variation in fossil fuel $CO_2$ emissions indicates a decreasing trend over the IdF region during the 2005-2021 period using the Mann-Kendall (MK) trend test at 5% level of significance (p-values ($0.0001$) $< 0.05$). According to the TNO 6km inventory, this is mainly linked

to the emission reductions in the residential and industry sectors, and further resulted from a decrease in coal use, an improvement in energy efficiency and building renovation. The inverse annual fossil fuel $CO_2$ emissions have been declining at a rate of ~2% per year over the four-year period from 2016 to 2019 (MK test p-values ($0.02$) $< 0.05$). In 2020, the COVID-19 pandemic and the continued levels of restrictions in place resulted in a dramatic decline in human activity. Our inversion results indicate a ~13% (12-14%) reduction in the annual emissions in 2020 compared to 2019, which is 2% (1-3%) larger than the prior estimate based on the

Origins.earth inventory (11%). The inverse annual emissions in 2021 rose by ~5.2% to $34.3\pm2.3$ $MtCO_2$ compared with 2020, but still remains 7.8% below the pre-COVID-19 level in 2019.

The aggregated annual fossil fuel $CO_2$ emissions obtained with the inversion are on average higher by ~5% for each year compared to the prior estimates from the Origins.earth inventory. Our inversion results are in agreement with the TNO 1km inventory in the whole city-scale fossil fuel $CO_2$ estimates for the years 2017 and 2018, while they are approximately 8.6%, 3.6% and 3.1% higher



than those estimated by Carbon Monitor Cities for the years 2019, 2020 and 2021, respectively. In 2018, the optimized emission is 38.1±2.6 MtCO$_2$. It is around 3.3, 2.7 and 0.8 MtCO$_2$ higher than that from AirParif, TNO 6 km and TNO 1km inventory respectively. Generally, the agreement among the various estimates of the annual fossil fuel CO$_2$ emission over the IdF region is within 10%, demonstrating robust emission estimates at the city scale through a combination of up-to-date inventories, atmospheric

modeling, and observations. It also provides evidence for a continuous and timely monitoring of urban fossil fuel CO$_2$ emission trends toward the reduction targets to achieve carbon neutrality.

**4 Conclusions and discussions**

This study shows the capacity of our city-scale inversion system, with a state-of-the-art inventory and high-precision continuous CO$_2$ measurements, for a timely and effective monitoring of urban fossil fuel CO$_2$ emissions over a long-term time period from

2016 to 2021. Our results indicate a decreasing trend in the annual CO$_2$ emissions over the IdF region with an amplitude of ~2% per year at 5% level of significance. The comparison of both prior and posterior annual emissions with independent estimates from other inventory datasets shows a less than 10% difference, which a satisfying target in terms of emission trend detection and verification for Paris to support its emissions-mitigation measures and related policy (Wu et al., 2016). In practice, few cities have such a high-resolution near-real-time local inventory like Paris. Through using the same annual total emission as a prior for the

year 2016-2018, the posterior emissions exhibit an emission reduction by ~3% over the IdF region when comparing the year 2016 with 2018. This demonstrates the ability of the inversion system to detect emission changes based on CO$_2$ measurements, independent of the information provided by the prior inventory. In addition to the afternoon CO$_2$ measurements that are commonly used in CO$_2$ inversion system, the assimilation of morning CO$_2$ data when the ABL heights are well reproduced by the model could lead to an additional 11~16% uncertainty reduction of the morning fossil fuel emissions. But it is worth pointing out that the

selection of morning CO$_2$ observations only for moderate to high wind speeds might reduce the observational constraint on the emissions in stable weather periods.

The uncertainties in the posterior estimates of CO$_2$ emissions are caused to a certain extent by errors in the spatio-temporal distribution of emissions at scales finer than the targeted ones. The present inversion system mainly controls the city-scale fossil fuel CO$_2$ emissions, but not the finer resolution in space. We thus further conducted a sensitivity inversion test using the TNO 1km

inventory as a priori as an alternative to the Origins.earth dataset for 2018. The other parameters were kept identical to the reference inversion configuration. The TNO inventory indicates more fossil fuel CO$_2$ emissions concentrated within the Paris inner city compared to the Origins.earth dataset (Figure S12). The inversion was able to spatially differentiate between the Paris city center and the Greater Paris region (excluding Paris), correcting the respective emissions differently (Figure S13). The posterior fossil fuel CO$_2$ emissions from the Paris city center were adjusted more upward with Origins.earth than TNO, while those from the rest

of Greater Paris region were corrected at a similar magnitude between Origins.earth and TNO. One can hope that a spatially explicit inversion system would allow to solve for the spatial distribution of urban emissions at the grid scale (Lauvaux et al., 2016). However, this will need additional information to determine the spatial correlation length of the inventory uncertainty or a high-density observation network to constrain the emissions from a large part of the city (Nalini et al., 2022).

Improvements in urban ecosystem modeling and monitoring for a precise accounting of urban biomass in the estimates of CO$_2$

fluxes is a relatively recent endeavor. Focusing on the Paris urban area, two limitations of this study have been acknowledged and are considered worthy of further investigation. Firstly, due to the coarse-resolution SYNMAP land use (1 km) data (Jung et al., 2006) and the MODIS satellite-derived vegetation indices (500m) used for the VPRM model, the simulated biogenic fluxes in Paris in this study are almost zero except for a few grid cells containing two big parks that are located in the eastern and western



outskirts of the Paris city respectively. While in reality, there are still a number of green space and pervious landscaped areas unevenly distributed in the city of Paris that need to be considered with a fine-scale (sub-kilometer) model. Secondly, there is a lack of detailed evaluation of the Paris-VPRM model since no eddy covariance measurement is available within Paris and its surroundings yet. Our analyses indicate that the actual biogenic fluxes within the IdF region may have a recognizable influence on

the measured $CO_2$ concentration gradients, whereas these biosphere signals are not well reproduced by the VPRM model. These discrepancies question the validity of the assumption that the signature of the local biogenic fluxes is not significant compared with that of the fossil fuel emissions in the measured gradients. Even though we have scaled up the prescribed observation errors to account for this possible large model bias in the biogenic fluxes and thereby to reduce its interference in the inverse fossil fuel emissions, the results may still be inevitably hampered. This study highlights the need for an in-depth analysis of the seasonal and

daily variations of the biogenic fluxes within the IdF region, especially during the cropland growing season. This could be achieved by improving the default VPRM model with a modified gross primary production and ecosystem respiration equations, domain-specific optimized parameters, high-resolution input datasets and high-quality diagnostic phenology[41]. In addition, measurements of carbon isotopes ([14]C, [13]C) and tracers co-emitted with $CO_2$ (e.g., CO, $NO_x$, VOCs) could also be used to separate the contributions from fossil fuel and biogenic components to the total $CO_2$ concentrations, which would be beneficial for the

optimization of sectoral $CO_2$ fluxes.

## Author contribution

JL, TL and PC designed the study. JL did the coding and implementation of the research. JL, TL, PC, FMB, GB, HU, MR and IA contributed to the analysis and interpretation of the results. MR, OL and MC coordinated scientifically the development of $CO_2$ measurement stations and ensured the calibration of the dataset. SK and MH processed the observed ABL data and gave precious

advice on the model-data evaluation. HU provided the Origins.earth inventory, OS and OP provided the AirParif emission estimates, HD and SD provided the TNO inventory, they all provided valuable feedback and opinions on the multiple dataset comparisons in section 3.3. JL prepared the manuscript with contributions and suggestions from all authors.

## Code/Data availability

The hourly averaged $CO_2$ data measured at 7 in situ stations are available on request from Michel Ramonet

(michel.ramonet@lsce.ipsl.fr).
The observed ABL height data are available on request from Simone Kotthaus (simone.kotthaus@ipsl.polytechnique.fr)
The Origins.earth $CO_2$ inventories are available on request from Hervé Utard (herve.utard@origins.earth).
The TNO $CO_2$ inventories are available on request from Hugo Anne Denier van der Gon (hugo.deniervandergon@tno.nl).
The ODIAC fossil fuel emission dataset was downloaded from https://db.cger.nies.go.jp/dataset/ODIAC/.

The Carbon Monitor Cities dataset is available at https://cities.carbonmonitor.org/.
The eddy covariance measurements were downloaded from the ICOS (https://doi.org/10.18160/2G60-ZHAK).

## Competing interests

The authors have no competing interests to declare.



**Acknowledgements**

The authors would like to thank SUEZ Group and La Ville de Paris for the support of this study. Thanks to Marc Jamous at CDS, to the IPSL QUALAIR platform team and to Cristelle Cailteau-Fischbach (LATMOS/IPSL) at JUS, and to LSCE/RAMCES technical staff for the maintenance of the $CO_2$ monitoring network. Thanks also to Pauline Buysse, Jérémie Depuydt, Daniel

Berveiller and Nicolas Delpierre for providing the eddy covariance flux measurements at Grignon and Fontainebleau (Barbeau) ecosystem sites. We thank Xiaobo Yang and Anna Agusti-Panareda from ECMWF for providing the near-real-time (NRT) CAMS $CO_2$ dataset. We also thank Elise Potier for her help in processing the TNO inventory. We thank the technical and IT staff at the SIRTA observatory for operation of the ALC. We thank Melania Van Hove and the AERIS-ESPRI data center for supporting the MLH product retrieval.

**Financial support**

Thomas Lauvaux was supported by the project CIUDAD (Make Our Planet Great Again program) and the Chair Fellowship CASAL (French Ministry of Research - CNRS).

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



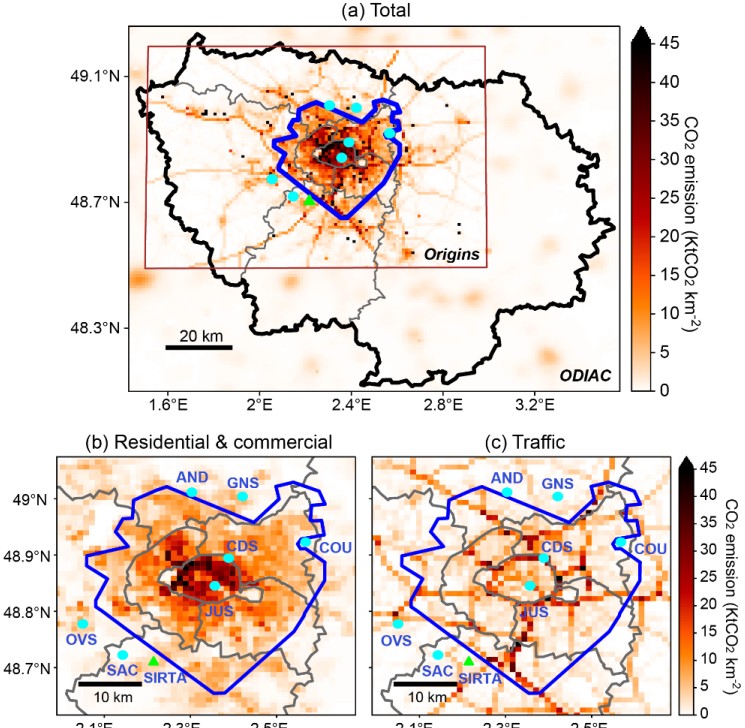

**Figure 1: Distributions of annual (a) total fossil fuel, (b) residential and commercial, and (c) traffic CO₂ emissions for the year 2019 according to the Origins.earth inventory (brown rectangle) complemented by the ODIAC dataset. The seven in situ CO₂ measurement stations are shown in cyan circles. The location of the SIRTA observatory with ABL measurements is marked by a green triangle. The black bold line shows the administrative limits of the Île-de-France (IdF) region. In the inversion system, emissions over two emitting regions are optimized: the Greater Paris region (within blue line) and the rest of the IdF region.**



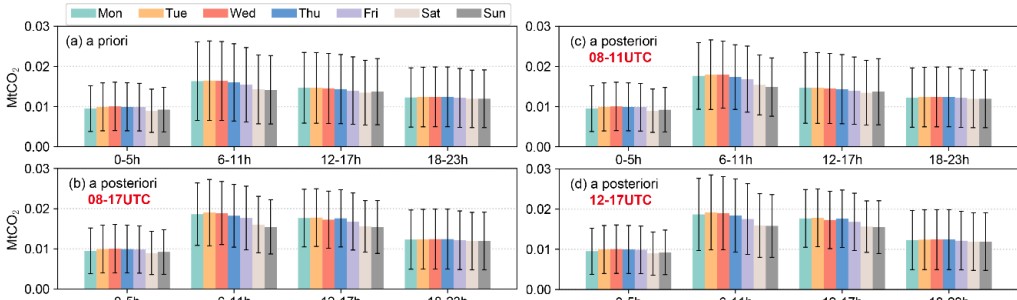

**Figure 2: Average (a) prior and (b-d) posterior fossil fuel CO₂ emission estimates and their uncertainties over the Greater Paris region over the year 2016-2021, for the four 6-hour time windows and the days of the week. (b-d) present the posterior CO₂ estimates when assimilating (b) daytime (8-17 UTC), (c) morning (8-11 UTC) and (d) afternoon (12-17 UTC) selected CO₂ concentration observations, respectively.**

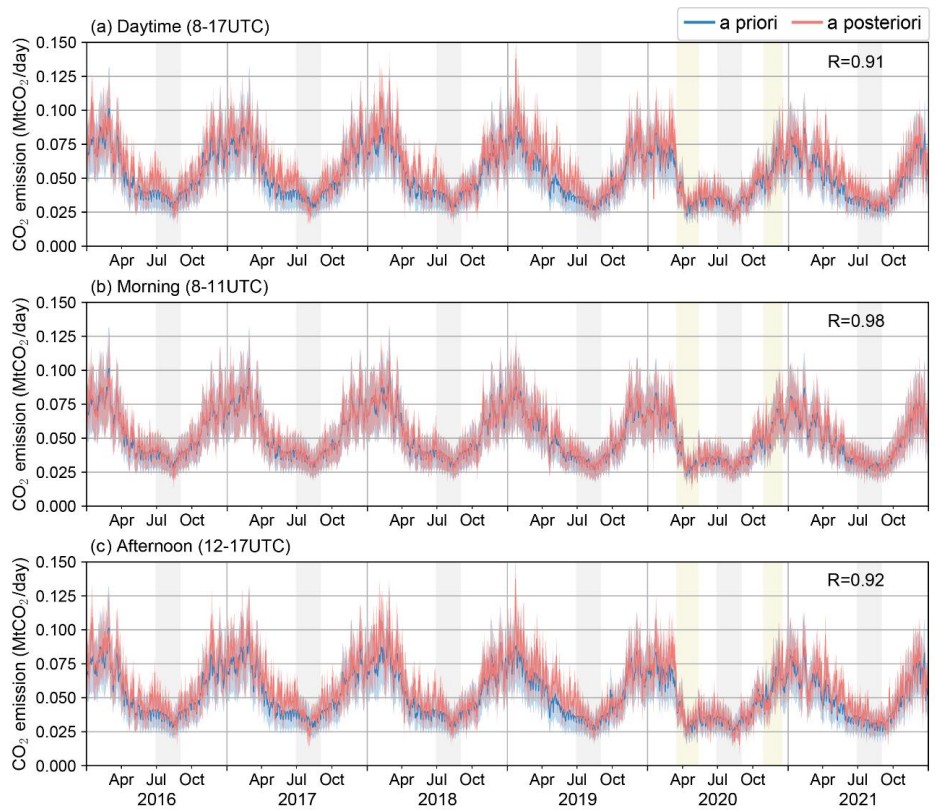

**Figure 3: Daily estimates of fossil fuel CO₂ emissions over the Greater Paris region when assimilating (a) daytime (8-17 UTC), (b) morning (8-11 UTC) and (c) afternoon (12-17 UTC) CO₂ concentration observations. The blue line and shading show the prior flux according to the Origins.earth inventory together with its assumed uncertainty. The pink and shading show the posterior estimates with their uncertainty ranges. The yellow shaded areas are the two COVID-19 lockdown periods in France. The grey shaded areas are the summer holidays of July and August.**



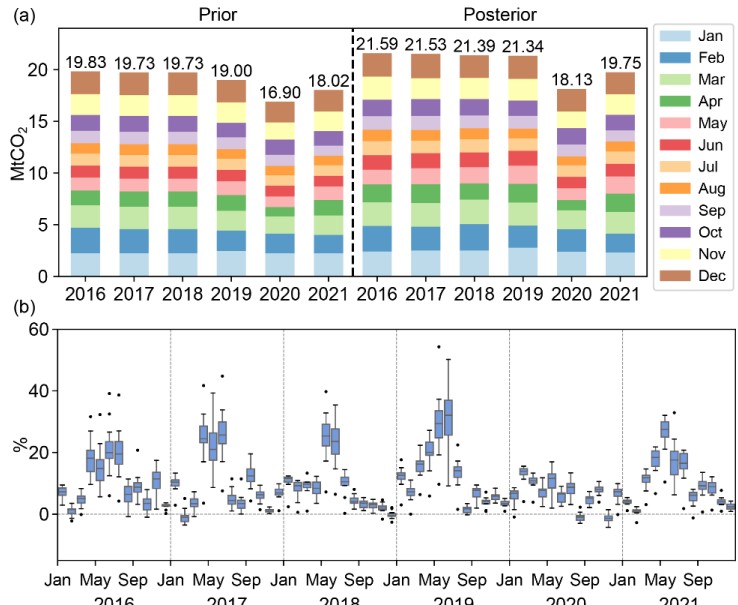

**Figure 4: (a) Prior and posterior estimates of the monthly total fossil fuel CO₂ emission over the Greater Paris region. (b) the change of CO₂ emissions in percentage (posterior-prior)/prior. The boxplots are the posterior emissions from an ensemble of sensitivity tests of the inversion configuration. Note that the prior emission for 2016 is slightly higher than 2017 and 2018 since it is a leap year.**

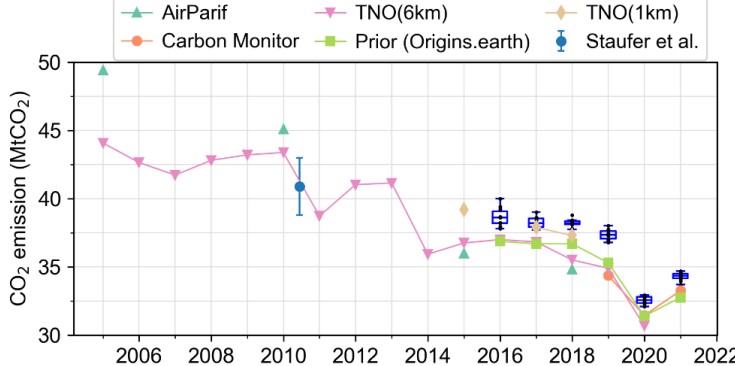

**Figure 5: Annual fossil fuel CO₂ emissions over the IdF region from 2005 to 2021. The blue boxplots present the distribution of posterior CO₂ emissions from an ensemble of sensitivity tests of the inversion configuration.**