# Peer review of "Can we use atmospheric CO2 measurements to verify emission trends reported by cities? Lessons from a six-year atmospheric inversion over Paris"

_EGUsphere, 2023_

## Author Comment (AC1)

We would like to thank the reviewer for the comments and suggestions to our manuscript. In the following, we answer to the reviewer's comments and indicate the changes in the manuscript that were implemented according to the recommendations. The comments are in black. Our answers are in blue.

**Referee #1:**

**General comments**

The manuscript presents results of a multiyear inverse modeling study estimating fossil $CO_2$ emissions using observations in the Paris region. The ability to estimate annual emissions with errors under 10% is a major advance presented in the paper. The paper is well-written and can be accepted after minor revision. One notable deficiency is lack of an inverse model description, suggest adding a section outlining the method.

**Response:**

We thank the referee for the positive comments on our manuscript. We have added <Section 2.4 Inversion configuration> together with Figure S4 and Table S1 in the revised manuscript to better clarify and summarize the atmospheric inverse modeling system used in this study.

**Detailed comments**

Page 1 Line 30 Suggest giving an uncertainty range to estimated 5.2% trend (seems to be in order of ±6% based on change of 32.6±2.2 $MtCO_2$ in 2020 to 34.3±2.3 $MtCO_2$ in 2021)

**Response:**

We have added an uncertainty range in both the abstract and section 3.3, respectively.

"Then, annual emissions increased by 5.2%±14.2% from 32.6±2.2 $MtCO_2$ in 2020 to 34.3±2.3 $MtCO_2$ in 2021."

"The inverse annual emissions in 2021 rose by about 5.2%±14.2% to 34.3±2.3 $MtCO_2$ compared with 2020 (32.6±2.2 $MtCO_2$), but still remains -8.0%±12.6% compared with the pre-COVID-19 level in 2019 (37.3±2.6 $MtCO_2$)."

P2 L18 Need to check if the references are most recent for Boston (Northeast corridor), and also for Los Angeles. Can mention dense NIST network around Washington DC.

**Response:**

Many top-down studies have used measurements of greenhouse gas mole fractions in an inverse modeling approach to estimate long-term (> 1 year) urban emissions. These studies have not only focused on $CO_2$ emissions as mentioned in the manuscript, but have also examined $CH_4$ emissions, such as the case of Los Angeles and Washington DC-Baltimore (Yadav et al., 2023; Karion et al., 2022). These investigations utilized observational data from various sources, including ground-based monitoring stations, aircraft measurements (e.g., Pitt et al., 2022), satellite (e.g., Lei et al., 2021) and more. We thus have refined the sentence to narrow down the scope and cited references that are more closely aligned with the statement. The modified text is as follows:

"To our knowledge, few estimates of city GHG emissions have been published when based on long-term tower-based measurements and atmospheric inversion systems. These include studies covering a period over one to five years for the cities of Paris, Boston, Indianapolis and Los Angeles (Staufer et al., 2016; Sargent et al., 2018; Lauvaux et al. 2020; Yadav et al., 2023)."

Meanwhile, as suggested, we have identified some more recent papers that are now cited as follows:

"The scientific capabilities evolve rapidly with increasing model performances (Deng et al.,

2017) and the deployment of dense networks in cities, e.g., Washington DC-Baltimore Metropolitan Areas (Karion et al., 2020), San Francisco Bay Area (Turner et al., 2020), Los Angeles (Yadav et al., 2021), Indianapolis (Davis et al., 2017), Paris, Munich and Zurich (https://www.icos-cp.eu/projects/icos-cities)."

P5 L17 Suggest giving readers more detail about the method used in Lian et al 2022, when reporting the revisions, that would save readers effort and help them understand the full merit of both this and the previous study. Also, need to give somewhere a summary of key points of the inversion approach, like, using station-to-station concentration gradients as "observations", control state, wind speed filters, horizontal resolution, PBL height filters, etc.

**Response:**

This suggestion is well taken. We have added section 2.4 in the revised manuscript to describe key points of the inversion approach, including 1) the setup of the control vectors, 2) the selection of the assimilated downwind-upwind $CO_2$ observation gradients (Figure S4), 3) the reference inversion setup and the sensitivity tests (Table S1), 4) two minor revisions compared to the configurations used in Lian et al. (2022).

P8 L10, L15 Better give 2% and 3% per year trend numbers with uncertainties like 2±X%.

**Response:**

Linear regression was conducted on the posterior emission data to derive a trend line for each inversion sensitivity test. Subsequently, the average trend and uncertainty were computed by considering all ten trend lines. We have added an uncertainty range for the 2% decrease trend in emissions from 2016 to 2021.

"Our results indicate a decreasing trend in the annual $CO_2$ emissions over the IdF region with an amplitude of ~2%±0.6% per year at 5% level of significance."

We have added an uncertainty range for the 3% decrease in emissions.

"the posterior emissions exhibit an emission change by about -3%±13.8% over the IdF region when comparing the year 2016 with 2018."

**Technical corrections**

P2 L10 Better write "inversion" instead of "atmospheric inversion" when citing Tarantola 2005.

**Response:**

Corrected.

P4 L4 For ODIAC, a more popular reference could be Oda et al ESSD 2018

**Response:**

Corrected.

P9 L12 Remove extra digits in text: diagnostic phenology41

**Response:**

Corrected.

Reference:

Lei, R., Feng, S., Danjou, A., Broquet, G., Wu, D., Lin, J. C., O'Dell, C. W., and Lauvaux, T.: Fossil fuel $CO_2$ emissions over metropolitan areas from space: A multi-model analysis of OCO-2 data over Lahore, Pakistan, Remote Sens. Environ., 264, 112625, https://doi.org/10.1016/j.rse.2021.112625, 2021.

Karion, A., Ghosh, S., Lopez-Coto, I., Mueller, K., Whetstone, J. R., & Pitt, J. R. Seasonal and Inter-annual Variability in Methane Emissions in Washington, DC and Baltimore, MD,

USA. In AGU Fall Meeting Abstracts (Vol. 2022, pp. A42L-04), December 2022.

Pitt, J. R., Lopez-Coto, I., Hajny, K. D., Tomlin, J., Kaeser, R., Jayarathne, T., Stirm, B. H., Floerchinger, C. R., Loughner, C. P., Gately, C. K., Hutyra, L. R., Gurney, K. R., Roest, G. S., Liang, J., Gourdji, S., Karion, A., Whetstone, J. R., and Shepson, P. B.: New York City greenhouse gas emissions estimated with inverse modeling of aircraft measurements, Elementa: Science of the Anthropocene, 10, https://doi.org/10.1525/elementa.2021.00082, 00082, 2022.

Yadav, V., Verhulst, K., Duren, R., Thorpe, A., Kim, J., Keeling, R., ... & Whetstone, J.: A declining trend of methane emissions in the Los Angeles basin from 2015 to 2020. Environmental Research Letters, 18(3), 034004, 2023.

---

## Author Comment (AC2)

We would like to thank the reviewer for the comments and suggestions to our manuscript. In the following, we answer to the reviewer's comments and indicate the changes in the manuscript that were implemented according to the recommendations. The comments are in black. Our answers are in blue.

**Referee #2:**

**Summary**

Lian et al. present a study investigating long-term changes in $CO_2$ emissions in the Greater Paris Area using different emission data products in combination with atmospheric observations. The inventories provided by origin.earth, AirParif and TNO can be the basis for policy decisions and they are validated using a bayesian inversion system which relies on assimilating morning and afternoon observations from a ground-based network. This emission monitoring framework performs well, is able to detect trends and short-term changes in emissions, here due to COVID-lockdowns. Overall, the paper is well-written and clearly structured. The description of the components is concise and a lot of information and illustrations of the actual performance are given in the appendix. The scope of the paper aligns very well with ACP and I can fully recommend publication after some minor changes have been considered.

**Response:**

We thank the referee for the positive comments on our manuscript.

**General comments**

1.) Unfortunately, the description of the modelling framework and its performance is very short. A lot of instructive and convincing information (plots) are only found in the supplemental materials. It could be worthwhile considering moving at least one into the main text.

**Response:**

This suggestion is well taken. We have added <Section 2.4 Inversion configuration> together with Figure S4 and Table S1 in the revised manuscript to better describe key points of the atmospheric inverse modeling system used in this study. In addition, we have also moved the figure <Monthly average daytime (8-17 UTC) observed $CO_2$ concentrations at seven in situ stations> into the main text as Figure 2.

2.) The manuscript does not discuss any other greenhouse gasses. In recent years several mobile surveys have been conducted highlighting significant $CH_4$ emissions in the region. It would be more balanced to mention $CH_4$, $N_2O$ as other gases that need to be mitigated (or why they can be ignored for the Plan Climat de Paris).

**Response:**

As given in the manuscript, the emission reduction targets in the Paris climate action plan refer to greenhouse gas (GHG) emissions, including not only $CO_2$ but also $CH_4$ and $N_2O$. It is worth noting that the emissions of $CH_4$ and $N_2O$ in the Paris region are much lower compared to those of $CO_2$ even when considering the global warming potential of these gases. According to the AirParif (official air quality agency of the Paris region, https://www.airparif.asso.fr/en/) inventory, the contribution of each GHG in $CO_2$ equivalent is 94% for $CO_2$, 4% for $N_2O$ and 2% for $CH_4$ in 2010 (AirParif, 2013). Defratyka et al. (2021) also reported that the natural gas network in Paris exhibited a leak rate of 0.11 leak indications per unique driven kilometer, which were classified as small leaks. Therefore, Paris is in the middle to low range compared to U.S cities, according to von Fischer et al. (2017) leak size categories. To highlight the emphasis of this study on $CO_2$ emissions while also addressing $CH_4$ and $N_2O$, we have added

the following sentence in the manuscript:

"According to the AirParif (official air quality agency of the Paris region, https://www.airparif.asso.fr/en/) inventory, the contribution of each of the main GHG (in term of $CO_2$ equivalent emission) is 94% for $CO_2$, 4% for $N_2O$ and 2% for $CH_4$ in 2010 (AirParif, 2013)."

**Specific comments**

P3L17: please provide a quantitative measure of the instrument performance, what does 'high precision' mean here?

**Response:**

According to previous studies in Paris (Xueref-Remy et al., 2018), when properly calibrated, the cavity ring-down spectroscopy (CRDS) instruments could have a high precision that is better than 0.1 ppm on hourly average $CO_2$ data. We have added the following sentence in the revised manuscript:

"The precision of the one-hour average $CO_2$ concentration is better than 0.1 ppm (Xueref-Remy et al., 2018)."

P4L33: formatting issue with "ru le"

**Response:**

Corrected.

P4L36: consider changing "imposed" to "required to be"

**Response:**

Text changed as suggested.

P15 L10: The blue boxplots should be added to the legend or the description of the other symbols to the captions. Splitting up the information seems unnecessary.

**Response:**

The blue boxplots have been added to the legend.

Reference:

AIRPARIF: Bilan des émissions de polluants atmosphériques et de gaz à effet de serre en Île-de-France pour l'année 2010 et historique 2000/2005, 2013.

Defratyka, S. M., Paris, J. D., Yver-Kwok, C., Fernandez, J. M., Korben, P., & Bousquet, P.: Mapping urban methane sources in Paris, France. Environmental science & technology, 55(13), 8583-8591, https://doi.org/10.1021/acs.est.1c00859, 2021.

von Fischer, J. C., Cooley, D., Chamberlain, S., Gaylord, A., Griebenow, C. J., Hamburg, S. P., ... & Ham, J.: Rapid, vehicle-based identification of location and magnitude of urban natural gas pipeline leaks. Environmental Science & Technology, 51(7), 4091-4099, https://doi.org/10.1021/acs.est.6b06095, 2017.

Xueref-Remy, I., Dieudonné, E., Vuillemin, C., Lopez, M., Lac, C., Schmidt, M., Delmotte, M., Chevallier, F., Ravetta, F., Perrussel, O., Ciais, P., Bréon, F.-M., Broquet, G., Ramonet, M., Spain, T. G., and Ampe, C.: Diurnal, synoptic and seasonal variability of atmospheric $CO_2$ in the Paris megacity area, Atmos. Chem. Phys., 18, 3335–3362, https://doi.org/10.5194/acp-18-3335-2018, 2018.